# Crack Propagation Analysis of Compression Loaded Rolling Elements

**DOI:** 10.3390/ma14102656

**Published:** 2021-05-19

**Authors:** Pavol Dlhý, Jan Poduška, Michael Berer, Anja Gosch, Ondrej Slávik, Luboš Náhlík, Pavel Hutař

**Affiliations:** 1Institute of Physics of Materials, Czech Academy of Sciences, Žižkova 22, 616 00 Brno, Czech Republic; pavol.dlhy@ceitec.vutbr.cz (P.D.); slavik@ipm.cz (O.S.); nahlik@ipm.cz (L.N.); hutar@ipm.cz (P.H.); 2Central European Institute of Technology, Brno University of Technology, Purkyňova 656/123, 612 00 Brno, Czech Republic; 3Faculty of Mechanical Engineering, Brno University of Technology, Technická 2896/2, 616 69 Brno, Czech Republic; 4Polymer Competence Center Leoben GmbH, Roseggerstrasse 12, 8700 Leoben, Austria; michael.berer@pccl.at; 5Material Science and Testing of Polymers, Montanuniversitaet Leoben, Otto Gloeckel-Straße 2, 8700 Leoben, Austria; anja.gosch@unileoben.ac.at

**Keywords:** fracture mechanics, crack propagation, stress intensity factor, thermoplastic material, bearing element, finite element method

## Abstract

The problem of crack propagation from internal defects in thermoplastic cylindrical bearing elements is addressed in this paper. The crack propagation in these elements takes place under mixed-mode conditions—i.e., all three possible loading modes (tensile opening mode I and shear opening modes II and III) of the crack are combined together. Moreover, their mutual relation changes during the rotation of the element. The dependency of the stress intensity factors on the crack length was described by general parametric equations. The model was then modified by adding a void to simulate the presence of a manufacturing defect. It was found that the influence of the void on the stress intensity factor values is quite high, but it fades with crack propagating further from the void. The effect of the friction between the crack faces was find negligible on stress intensity factor values. The results presented in this paper can be directly used for the calculation of bearing elements lifetime without complicated finite element simulations.

## 1. Introduction

It has become quite common nowadays to use thermoplastic materials for the production of cylindrical rollers in roller bearings or linear guidance parts. Thermoplastics are often preferred for these applications because of their light weight and effective mass production [1]. Typical materials for these applications are polyamide (PA), polyoxymethylene (POM) and polyetheretherketone (PEEK) [1,2,3,4,5].

The cylindrical bearing elements are subjected to rolling in their operation. Parts subjected to rolling often fail due to cracks initiating on the surface (or slightly under the surface) that are in contact with another part (typically a metal rail, linear guide or a bearing ring). The mechanism is called rolling contact fatigue and it is usually associated with metal wheels running on rails [6,7,8,9], but it appears also in bearing elements such as balls and cylinders [10,11,12,13]. It is important to note that in the rolling contact fatigue the cracks grow under mixed-mode loading conditions, which means the shear modes II (in-plane shear) and III (out-of-plane shear) play a significant role in the crack propagation.

There is also another mechanism of failure of the bearing elements—the crack propagates from an internal defect until it reaches a critical length and the bearing element breaks. This mechanism is important for thermoplastic bearing elements, especially cylinders. Thermoplastic cylindrical bearing elements are typically manufactured by injection molding, which is a very common way well known for its high processing speed. However, internal defects occur in the cylinders as a result of significant shrinkage from the solidification of the molten polymer in the mold after the injection [3]. When a defected cylinder is installed in a bearing and starts rolling under load, a crack can initiate from the shrinkage defect and start propagating. This process eventually results in failure of the cylinder. Similar to the previously mentioned rolling contact fatigue process, the crack grows under mixed-mode conditions that are caused by the ever-changing orientation of the crack to the load acting during rolling. As the production defects are very difficult to eliminate [3], it is important to describe the process that leads to failure and thus enable reasonable lifetime estimation of these parts.

The crack propagation in this case can be described by the linear elastic fracture mechanics (LEFM) parameters, e.g., the stress intensity factors. The relation between fatigue crack growth rate and stress intensity factor for polymer materials is commonly used [14,15,16,17,18,19,20,21,22]. Then, an estimation of the total lifetime can be performed if the crack growth rate is known from experiments and the considered crack is accurately described by fracture parameters—see details in [23]. This process is well established and used, e.g., for polyethylene (PE) pipes [23,24,25,26], or recently for polymer parts produced by additive manufacturing [27,28]. Approaches for lifetime estimation based on fracture mechanics are used also for the delamination in composites with polymer matrix [29] and plastic-encapsulated microelectronics [30]. However, all the referenced studies considered mode I crack propagation only. Frequent presence of mixed-mode conditions in fracture mechanics problems concerning parts such as bearing elements or gears made of thermoplastics such as POM, PEEK and other is fueling recent interest in testing the crack growth conditions in these materials in mixed-mode conditions—see [31]. It was found that mixed-mode loading conditions lead to a clear lifetime reduction in the case of mode I/mode III loading. Additionally, during fatigue testing in mixed-mode conditions important temperature increase, caused by friction between crack flanks, was observed. This temperature increase depends on the loading frequency and can significantly deteriorate mechanical properties of the polymer. Therefore, any lifetime prediction that neglects the effect of the mixed-mode conditions leads to strongly non-conservative results.

The detailed knowledge of the mode mixity and accurate estimation of the stress intensity factors during crack propagation is necessary for further lifetime prediction of the component. A thorough finite element simulation (FEM) of a real component where cracks propagate in mixed-mode conditions is also essential for the further practical application of the findings of previous research on the mixed-mode crack propagation in thermoplastics [31,32,33].

Detailed FEM simulation of the crack propagation in cylindrical bearing elements made of POM during rolling is described in this paper. This study is a direct follow-up to the work of Berer et al. [3], but it takes into account the presence of the mixed-mode conditions at the crack front. The stress intensity factors for the mode I (tensile crack opening), mode II (in-plane shear crack opening) and mode III (out-of-plane shear crack opening) were determined and their mutual relationship was assessed. General equations that allow the assessment of stress intensity factors for a wide range of dimensions of the cylinders were established. Different conditions of the crack propagation were considered in the simulation. Attention was paid to the shape of the initial defect and the friction between crack faces, as they come into contact in some positions of the rolling cylinder.

## 2. Numerical Models

A parametrical numerical model was developed in order to simulate a growing crack inside a rolling cylinder under different conditions and to determine fracture mechanics parameters—the stress intensity factors. The software ANSYS was used for the simulation. The model was created using a parametric macro programmed in the APDL (ANSYS Parametric Design Language).

Three different modifications of the model, that differ in some features influencing the crack growth, were created:Basic model with a flat crack—model of a cylinder compressed between two steel plates, that contains one circular flat crack in the middle. Although the faces of the crack can come in contact, no friction is considered between the faces. Different combinations of dimensions were considered.Model with a void—the crack in this model is not flat from the start, but it starts from a three-dimensional void in the middle of the cylinder. Although it starts from the void, the crack is still considered flat and circular.Model with a void and friction—the same as the model with a void, but friction is considered between the faces of the crack.

The details of these models are described further in this section. The results are discussed in Section 3.

### 2.1. Basic Model with a Flat Crack

The most important part of the model is a cylinder with a flat circular crack in the middle (the crack is modelled directly as a part of the geometry, refer to ANSYS APDL Help for more details about crack analysis [34]). Apart from the cylinder, two steel plates, one at the top and one at the bottom, compressing the cylinder were added to model the case of a cylinder installed between rails or bearing rings. The main dimensions of the basic model cylinder are denoted *D* (diameter) and *L* (length). The crack length is denoted *a*. The model is parametric, which means it is possible to simulate almost any set of dimensions and also the change in orientation of the crack during rotation of the cylinder. See the modelled situation in Figure 1.

Only one half of the cylinder was modelled to take advantage of the symmetry and reduce the number of elements and computing time. Further reduction of the model was not possible, because other planes of symmetry are disrupted by the rotation of the crack inside the cylinder. Quadratic tetrahedral elements were used to mesh the most of the model. The mesh was made very fine and a little more regular around the crack front—quadratic brick mesh was used in the area surrounding the crack front with special quarter-point elements directly at the crack front (see the mesh of the model with detailed view of the crack front in Figure 2). The material model of the cylinder was a linear elastic, isotropic solid defined by the Young’s modulus of 3.6 GPa and Poisson’s ratio of 0.45. These values are typical for the POM used to make bearing elements.

The steel plates compressing the cylinder at the top and at the bottom were meshed by regular quadratic bricks. The material model of these parts was linear elastic, isotropic, with Young’s modulus of 210 GPa and Poisson’s ratio of 0.3. Mutual contact was defined between the steel plates and the cylinder.

Boundary conditions were defined for the whole model. Symmetrical boundary conditions were defined on the plane of symmetry. Fixed support was defined at the bottom of the lower steel plate. The upper steel plate was loaded by the force of 350 N. The boundary conditions and loads are schematically depicted in Figure 2.

The same basic set of 7 combinations of dimensions *D* × *L* as in Berer et al.’s previous work [3] was chosen to be modelled (dimensions are in mm): 3 × 3, 4 × 4, 5 × 5, 6 × 6, 6 × 3, 6 × 9, 6 × 12. Nine more combinations were added to make the set more robust for the intended parametric study: 3 × 4, 3 × 5, 3 × 6, 3 × 9, 3 × 12, 4 × 5, 4 × 6, 5 × 4 and 6 × 4.

The growth of the crack inside the cylinder was simulated by modelling the cylinder with the crack length *a* going from 0.25 mm up to 1.75 mm by the steps of 0.50 mm with the addition of two extra crack lengths of 0.85 mm and 1.00 mm (this was done for a better comparison of the models with and without a void inside, which is discussed further in this paper). Note that, in this case, the crack is circular and the crack length *a* is its radius (as shown in Figure 1).

For every crack length, the stress intensity factors *K_I_*, *K_II_* and *K_III_* were determined along the whole crack front. The position on the crack front is defined by the angle *γ* that goes from +90° across 0° to −90° (see Figure 1 for illustration).

The cylinder was considered rolling between the two compressing steel plates to simulate the operation of a bearing element. The rolling was achieved by changing the orientation of the crack in the model. It is important to remark, that contact was defined also between the faces of the crack, because the crack faces would be pressed against each other in some of the positions. The orientation of the crack is defined by the angle *ρ* between the plane of the crack and the vertical axis of the model (or the direction of forces acting on the steel plates, see Figure 3). The angle *ρ* goes from 0° up to 180° by the steps of 15°. For a better description of the change in the values during rolling, eight steps of the angle *ρ* were added—3.8°, 7.5°, 11.2° and 22.5° as well as 157.5°, 168.8°, 172.5° and 176.2. The range 0°–180° simulates only a half turn of the cylinder, but the remaining part of the turn would be symmetrical. The crack growth from 0.25 to 1.75 mm and the stress intensity factor determination was carried out in every step of the rolling. More than 400 simulations in total were carried out.

In every step of the crack propagation (and in every version of the model), the stress intensity factors were evaluated using two different methods. The first method was the domain integral [35], which is fully integrated in the software ANSYS. The second method was the determination of stress intensity factors directly from the deformation of the nodes of the special crack tip elements [36].

### 2.2. Models with a Void and Friction

The models with the circular flat crack give an idea about the crack propagation and the mutual ratios of the crack propagation modes. However, the flat crack does not exactly reflect the reality of the bearing cylinders. The shrinkage defects, from which the cracks usually start, are not just simple flat discontinuities in the material. This was the reason to modify the basic model with a flat crack and create a model that contains a void in the middle. In the case of the models with a void and friction, the basic dimensions *D* × *L* were kept constant at 6 × 6 mm^2^. The variable parameters were the crack length, the orientation of the crack and the dimensions of the void.

In practice, the void was observed to be a rather complex three-dimensional, spheroidal structure with a strong variability in shape and surface roughness within the same class of bearing elements. Moreover, size, shape and surface topology depend significantly on processing parameters and the processed polymers. Hence, to keep it on a useful complexity level and in order to obtain generalizable data, the void was represented by a regular spheroid. The radius of the void *r_v_* was 0.75 mm (diameter *d_v_* = 1.5 mm). The height of the void *h_v_* was set as a variable to study its effect. See Figure 4 for the illustration of the situation.

To see what different shapes of the voids do to the stress distribution around the crack, the height of the void was varied in the model. The radius of the void was kept constant. The void height is described here in terms of a ratio of the height to the radius (further referred to as the void ratio)—*h_v_/r_v_*. The void ratio values were considered going from 0.125 up to 1.75 (with the step of 0.25 between 0.25 and 1.75). The void is shaped such as a perfect sphere in case of *h_v_/r_v_* = 1. In case of *h_v_/r_v_* < 1, the void is an oblate spheroid; in case of *h_v_/r_v_* > 1, the void has a shape of a prolate spheroid. Apart from the void, the model contains a circular crack similar to the model with a flat crack. The crack lengths *a* considered in the model with the void were the same as in the model with a flat crack, except the 0.25 mm and 0.75 mm. It was not possible to model these lengths due to the void in the middle. The crack length started at 0.85 mm and went up to 1.75 mm.

The model with the void was further modified by adding friction between the contact faces of the crack. It was assumed that the friction could slightly change the shear stress distribution in some of the crack positions, which could have an effect on the stress intensity factors *K_II_* and *K_III_*. The coefficient of friction used in the contact was 0.32. This value was chosen as a conservative estimate based on relevant data about friction coefficients of POM and PEEK materials, which are usually peeking slightly above 0.3 at room temperature [37,38,39,40]. However, the friction coefficient can significantly vary for different blends of the same polymer.

## 3. Results and Discussion

At first, the stress intensity factors determined by the model with a flat crack were compared to values published by Berer et al. in [3] to validate the functionality of the new model described in the previous section. The numerical model by Berer et al. represents one eighth of the bearing cylinder and it was not used to determine the values of stress intensity factors *K_II_* and *_KIII_*, only *K_I_* was evaluated. The values for the case of the cylinder with *D* × *L* = 6 × 6 mm^2^ in the rolling position *ρ* = 0° were taken from the paper and compared to the results that were produced by the model described here. The comparison is plotted in Figure 5. The values of *K_I_* are plotted as a function of the position on the crack front described by the angle *γ*. The discrepancy between the results of the two models is negligible. Note that Berer et al.’s values are in the range of 0°–90°, because the one-eighth-type of symmetry was used in their model. It was possible, because the values of *K_I_* are symmetrical, as shown by the newer results. However, it would not be possible to use this type of symmetry for the evaluation of *K_II_* and *K_III_*.

It was mentioned above, that the stress intensity factor values were evaluated by two different methods in the model described here. The results produced by both these methods were also compared and they are plotted in Figure 5. The solid line in Figure 5 represents the domain integral method; the separate points (in the color of the line) represent the estimation from the node deformations. There is a nice match between the values produced by the two different methods, which also validates the functionality of the model. All the values presented further in this paper were obtained by the domain integral method.

In the following sections of this paper, results from the 3 versions of the model are presented and discussed.

### 3.1. Model with a Flat Crack—Results and Discussion

It was mentioned above that the orientation of the crack to the acting load changes due to the rolling of the cylinder. The pure mode I appears only in a few instances during the turn. There is always rather a combination of modes I, II and III. The modes and their ratio change depending on the position of the cylinder (angle *ρ*) and they are also different for different positions on the crack front (given by the angle *γ*).

The results of the basic model with a flat crack provide an idea on how the modes change during one turn of the cylinder. The following figures contain plotted results for the cylinder with the dimensions of *D* × *L* = 6 × 6 mm^2^ and loading force *F* = 350 N, unless stated otherwise. The results are plotted in Figure 6 (*K_I_*), Figure 7 (*K_II_*) and Figure 8 (*K_III_*) for one of the simulated crack lengths—1.25 mm. 3D plots were chosen to visualize the values of stress intensity factors depending on both, the position on the crack front *γ* and the overall rolling orientation *ρ* of the crack. The results were also plotted in the form of 2D plots, where the crack length *a* and position on the crack front *γ* were fixed and the stress intensity factors are plotted as a function of the rolling orientation (angle *ρ*). Many 2D plots had to be created to illustrate the whole situation, because of many possible combinations of parameters (crack length and position on the crack front). The 2D plots are not included in the text of this paper for the sake of clarity. The most important 2D plots are included in Appendix A.

The 3D plot of *K_I_* (Figure 6) shows that the *K_I_* values reach their maximum at the beginning and at the end of the turn of the cylinder. During the turn from 0° towards 180°, the values decrease up to the point when the two crack faces come into contact. The crack stays closed until the cylinder comes into a position where the opening stress starts acting on the crack again. The *K_I_* values are zero when the crack faces are in contact in the model. A simulation without the contact of crack faces was also carried out to investigate the exact moment of the crack closing and opening. If no contact is defined between the crack faces, the *K_I_* values become negative in the part of the cycle where the crack is closed (the negative values are also plotted in Figure 6). Negative values of *K_I_* cannot occur in reality, but this kind of simulation helps to evaluate the cycle of *K_I_* and its asymmetry, which can be important for a later use in lifetime estimations and for experimental testing of such a situation.

The points of crack closing and opening are slightly different for different positions on the crack front—for the plotted crack length of 1.25 mm, the crack closes a little earlier for *γ* = +90° (and −90°) than for *γ* = 0° (this can be better observed in Figure A5 in Appendix A). Their position also depends on the current crack length (the shift in the position can be observed in Figure A1, Figure A2, Figure A3, Figure A4, Figure A5 and Figure A6 in Appendix A), which suggests that the crack might not propagate exactly as a regular circle in reality.

The values of stress intensity factors for the shear modes, the *K_II_* and *K_III_*, are higher in magnitude compared to the *K_I_* in terms of maximum values. On the beginning of the turn, both *K_II_* and *K_III_* are zero along the whole crack front, as the crack is not subjected to shear loading at all. However, the conditions change with the turning. The middle of the crack (*γ* = 0°) is subjected to mode III type of loading and the mode II does not appear here at all during the turn, whereas the crack front ends (*γ* = 90° and −90°) develop mutually opposite values of *K_II_* during the cycle, and *K_III_* remains equal to zero. In between these positions, the *K_II_* and *K_III_* values follow different sine patterns. As the crack becomes perpendicular to the direction of loading (*ρ* = 90°), both of the shear modes disappear again. Then, in the following part of the turn, the values of *K_II_* and *K_III_* appear again in the same places on the crack front, but with opposite signs—see Figure 7 and Figure 8.

The crack tip mixed-mode loading has a completely out-of-phase character. Again, refer to Figure A1, Figure A2, Figure A3, Figure A4, Figure A5 and Figure A6 in Appendix A for a more detailed view on the changes of K-values during the turn. When the *K_I_* reaches the maximum values, both *K_II_* and *K_III_* are zero. There are short intervals at the beginning and the end of the turn, when the crack is open and loaded by a combination of *K_I_* and *K_III_* (for the position of *γ* = 0°) or *K_I_* and *K_II_* (for *γ* = 90° and −90°). Between these, there are intervals in which the *K_II_* and *K_III_* are non-zero and even reach their maxima (or minima) and the crack is closed (*K_I_* is zero). This means that the crack faces are being forced against each other and into one of the shear modes at the same time. It is quite likely that heat is generated by the friction of the crack faces in these intervals, which can have an influence on the crack propagation rate [31,32,41].

The instant values of *K* are not very practical for the description of the crack growth kinetics in the investigated situation. It is more practical to use the maximum values of stress intensity factor *K_max_* to describe the whole cycle. In Figure 9, these values are plotted as a function of the normalized crack length *a/W* (where *a* is the crack length and *W* corresponds to the radius or half of the length of the whole cylinder depending on position *γ* = 90° (−90°) or *γ* = 0°, respectively (see Figure 1). It is important to note here, that the rolling position *ρ*, at which the maxima and minima of *K_I_* and *K_III_* are reached, are constant with the growing crack length *a*. The maximum of *K_I_* can be always found at *ρ* = 0° and *ρ* = 180°, the (theoretical) minimum at *ρ* = 90°. The maximum of *K_III_* stays at *ρ* = 135° and minimum at *ρ* = 45°. However, the position where the *K_II_* reaches its extreme (minimum or maximum depending on the position, if *γ =* 90° or −90°) gradually shifts from 45° towards lower values of *ρ* with the crack length increasing and similarly the position of the other extreme shifts from 135° towards higher values. The shift can be observed in Figure A1, Figure A2, Figure A3, Figure A4, Figure A5 and Figure A6 in the Appendix A. The cause of this is most likely that the crack becomes more influenced by a complicated stress state in the vicinity of the contact with the loading plates, which manifests itself the most at the positions *γ* = 90° and −90°, where *K_II_* reaches its maxima and minima. This shift in the position does not influence the characterization of the stress intensity factor cycles using the *K_max_* values though.

However, the magnitude of the stress intensity factor itself does not provide information about the character of the cycle. For the correct description of the loading cycle, it is necessary to specify also the asymmetry of the cycle in terms of *R*-ratio. The *R*-ratio is the ratio of the minimum value *K_min_* to the maximum value *K_max_* of the cycle.

In the investigated case, the *R*-ratio is dependent on the position at the crack front and also on the current crack length. The dependency is plotted in Figure 10. In case of mode I, the *R*-ratio is negative (this means that the minimum value is negative and the maximum is positive); it is lower in the 0° position on the crack front, where the minimum has a lower value, and it gets slightly higher towards the 90° (and −90°) position. The ratios in all the positions are rising with the growing crack. Additionally, the difference between the positions becomes more pronounced with the crack growing larger. The reason probably is that the position of *γ* = 90° (−90°) starts to be more influenced by the stress distribution caused by the contact zone of bearing element and steel plates, when the crack length *a* grows larger. In case of mode II, the *R*-ratio of −1 (this denotes a symmetrical cycle) is the same for every position on the crack front, apart from the 0°, where the *K_II_* is 0. Analogically to the mode II, the *R*-ratio for the mode III is equal to −1, apart from the 90°and −90° positions.

The *K_max_* values were determined for more combinations of dimensions *D* × *L*. The considered diameters *D* were 3, 4, 5, 6 mm, the considered lengths *L* were 3, 4, 5, 6, 9 and 12 mm. The entire range of crack lengths, as it is specified in Section 2, was considered for the 6 × 6 mm^2^ type of cylinder only. Only some crack lengths were considered for the other combinations of length and diameter. These crack lengths were chosen with respect to the dimensions of the particular combination, because some of the lengths did not fit in the particular combination.

The obtained values of *K_max_* were then fitted with parametric functions that define the dependency of the stress intensity factors on the crack length and are generalized with respect to the loading force and dimensions of the cylinder. The fits were carried out for every point on the crack face, where the stress intensity factor reaches its maximum during the turn—this means there is one fit for the *K_IImax_* in the *γ* = 90° position and one for the *K_IIImax_* in the *γ* = 0° position. Two fits for the *K_Imax_* were made, one for the 90° position and another for the 0° position, because the difference between the maxima in these two positions is not very pronounced, although technically the global maximum is reached only in the 0° position. The position *γ* on the crack front is indicated by respective indices.

The functions are slightly different for every type of the stress intensity factor, but they always feature the loading force *F*, the square route of the crack length *a*, one of the dimensions of the cylinder (either the diameter *D* or the length *L*) and a dimensionless shape function *Y* that was found by curve fitting.

The shape function curve fits for the *K_Imax_*
_90°_ and *K_Imax_*
_0°_ are plotted in Figure 11a,b, respectively. The equation describing the dependencies are the following:(1)$$K_{Imax\;90{}^{\circ}}=\frac{F\;}{\sqrt{10^{3}}\sqrt{a}\;L}Y_{Imax\;90{}^{\circ}}\left(\frac{2a}{D}\right),$$
(2)$$K_{Imax\;0{}^{\circ}}=\frac{F\;}{\sqrt{10^{3}}\sqrt{a}\;D}Y_{Imax\;0{}^{\circ}}\left(\frac{2a}{L}\right).$$
where *F* is the loading force in N, *a* is the crack length in mm, *D* is the cylinder diameter in mm, *L* is the cylinder length in mm and *Y_I_*
_90°_ and *Y_I_*
_0°_ are the dimensionless shape functions that have been found in the following form:(3)$$Y_{Imax\;90{}^{\circ}}\left(\frac{2a}{D}\right)=0.389\left(\frac{2a}{D}\right)-0.014,$$
(4)$$Y_{Imax\;0{}^{\circ}}\left(\frac{2a}{L}\right)=0.354\left(\frac{2a}{L}\right)-0.004.$$

Even though the dimensions are in mm, the resultant unit of the stress intensity factors calculated using Equations (1) and (2) is in MPa·m^1/2^, which is the typical unit used for the stress intensity factors. This is ensured by the factor of $\sqrt{10^{3}}$ in the denominator of both equations.

The scatter of the points around the line in Figure 11 determines the level of generalization, which is achieved by the fit, and the difference between the parametric functions and the actual determined values. In cases of *Y_Imax_*
_0°_ and *Y_Imax_*
_90°_, the difference is usually not more than 6%, but for some combinations it goes up to 30% (especially when there is a larger difference between *D* and *L* of the cylinder).

To describe the cycle of *K_I_* properly, the *K_Imin_* values are also needed, because the *R*-ratio does not stay constant for the *K_I_* cycle during the crack propagation (as illustrated in Figure 10). Even though the *K_Imin_* values are only theoretical, because in practice the crack is closed and the stress intensity factor is equal to zero, knowing these values makes it possible to describe the whole cycle in detail and most importantly to determine the precise moments of the crack closing and opening. The *K_Imin_* functions were created in the same manner as the *K_Imax_* functions above. The shape function fits are plotted in Figure 12a,b. The equations follow:(5)$$K_{Imin\;90{}^{\circ}}=\frac{F\;}{\sqrt{10^{3}}\sqrt{a}\;L}Y_{Imin\;90{}^{\circ}}\left(\frac{2a}{D}\right),$$
(6)$$K_{Imin\;0{}^{\circ}}=\frac{F\;}{\sqrt{10^{3}}\sqrt{a}\;D}Y_{Imin\;0{}^{\circ}}\left(\frac{2a}{L}\right).$$

The *Y_Imin_*
_90°_ and *Y_Imin_*
_0°_ are the dimensionless shape functions that have been found in the following form:(7)$$Y_{Imin\;90{}^{\circ}}\left(\frac{2a}{D}\right)=-0.714\left(\frac{2a}{D}\right)-0.053,$$
(8)$$Y_{Imin\;0{}^{\circ}}\left(\frac{2a}{L}\right)=-1.106\left(\frac{2a}{L}\right)+0.008.$$

The parametric functions were also found for the *K_IImax 90°_* and *K_IIImax 0°_* values. The fits for these values are plotted in Figure 13a,b, respectively. The generalized *K_IImax_* and *K_IIImax_* are much less scattered, which means that the parametric function provides a very good estimation of the real stress intensity factor values. The parametric functions have a similar form to the previous functions of *K_Imax_*. The equations are the following:(9)$$K_{IImax\;90{}^{\circ}}=\frac{F\;}{\sqrt{10^{3}}\sqrt{a}\;L}Y_{IImax\;90{}^{\circ}}\left(\frac{2a}{D}\right),$$
(10)$$K_{IIImax\;0{}^{\circ}}=\frac{F\;}{\sqrt{10^{3}}\sqrt{a}\;D}Y_{IIImax\;0{}^{\circ}}\left(\frac{2a}{L}\right).$$

The equations describing functions *Y_II_*
_90°_ and *Y_III_*
_0°_ were found to be the following:(11)$$Y_{IImax\;90{}^{\circ}}\left(\frac{2a}{D}\right)=0.884\left(\frac{2a}{D}\right)-0.012,$$
(12)$$Y_{IIImax\;0{}^{\circ}}\left(\frac{2a}{L}\right)=0.512\left(\frac{2a}{L}\right)-0.002.$$

Again, all the input dimensions in Equations (5)–(8) are in mm and the load *F* in N. The resultant unit of the stress intensity factor is MPa·m^1/2^.

All of the above stated equations are valid only in the range of 0 < *2a*/*D* < 0.6 and 0 < *2a*/*L* < 0.6, respectively, because the stress intensity factors were not evaluated outside this range. Care must be taken also if using the equations for a similar situation with different material parameters. Although the material parameters do not influence the stress intensity factors directly, they can influence the contact zone. This means that stress intensity values for larger crack lengths *a* may be influenced by the material parameters. However, the rest of the cases will not be influenced by a change in the material parameters, because the stress intensity factors in these cases only depend on the loading force and dimensions of the cylinders.

### 3.2. Models with Voids and Friction—Results and Discussion

This section summarizes the results from those versions of the model that contained an idealized void. All of the models with the voids were cylinders with *D* × *L* = 6 × 6 mm^2^ and with a spheroidal void in the middle. The radius of the void *r_v_* was 0.75 mm and the height of the void *h_v_* was variable. In the following, the void dimensions are described by the void ratio *h_v_/r_v_*.

The general influence of the dimensions of the void on the stress intensity factors is illustrated by the results of a model with constant crack length of 0.85 mm and different void heights *h_v_*. Figure 14a shows the *K_Imax_* values determined for the crack front positions *γ* = 90° (−90°) and *γ* = 0°. For the void ratio (*h_v_/r_v_*) of 0.125, the *K_Imax_* values in both *γ* positions are the same as for the flat crack with the same crack length (*a* = 0.85 mm). After that, *K_Imax_*
_90°_ increases with increasing void ratio until the ratio of 1.75, where it is approximately three times higher. *K_Imax_*
_0°_ on the other hand, decreases to lower values than those calculated for the flat crack until it becomes zero. This is caused by the decrease in stiffness of the cylinder, when the void gets more prolate (*h_v_/r_v_* > 1)—the crack then opens more in the positions of 90° and −90°, and less in the 0° position. Additionally, the values of stress intensity factors in the positions of 90° and −90° on the crack front are heavily influenced by the contact zone stress field.

The value of *K_Imax_*
_90°,0°_ for the case of a flat crack (without the void) of the same length *a* = 0.85 mm is plotted as a solid line for comparison. Plotting this value as a solid line was chosen for the plot to be clearer, but it is technically incorrect, because the value would normally show as a point at *r_v_/h_v_* = 0.

The values of *K_IImax_* for the *γ* = 90° (−90°) position and the *K_IIImax_* values for the *γ* = 0° position are shown in Figure 14b. The values of *K_IImax_*
_90°_ and *K_IIImax_*
_0°_ for the case of a flat crack of the same length *a* = 0.85 mm are plotted as solid lines for comparison. The *K_IImax_*
_90°_ values for the void ratio of 0.125 are slightly lower than the *K_IImax_*
_90°_ values for the flat crack length with the same crack length—by about 11%. With increasing void ratio, the *K_IImax_*
_90°_ values decrease quite rapidly. For the void ratio of 1.75, they are approximately seven times lower than the values without the void. This is exactly the opposite tendency compared to K*_Imax_*
_90°_ in the same *γ* position. *K_IIImax_*
_0°_ shows only a weak dependency on the void ratio. The *K_IIImax_*
_0°_ for the void ratio of 0.125 is the same as for the flat crack. For the highest simulated void ratio of 1.75, the *K_IIImax_*
_0°_ values are only slightly higher than those for the same flat crack length (by about 10%).

The results above suggest a strong influence of the void on the stress distribution around the crack tip (described by stress intensity factors). To see how the change in the void dimensions influences the actual crack propagation, crack propagation with the void was simulated. The void ratio of 0.25 (oblate spheroid) and 1 (perfect sphere) were simulated—these were chosen arbitrarily so that it was possible to describe a trend in the behavior. As was mentioned above, the crack growth from 0.85 mm up to 1.75 mm was considered.

Figure 15 shows the values of *K_max_* from the flat crack model compared to the values from the models with the voids. The results agree with previous observations on the dependencies on the varying void ratio. The presence of the void increases the *K_I_* in the 90° position and decreases the *K_I_* in the 0° position. The *K_II_* is decreased substantially by the void, whereas *K_III_* is subject to a mild increase. One feature in the results is common for all the stress intensity factors (no matter the loading mode nor the positions on the crack front): with the crack propagating further from the void, the stress intensity factors are less influenced by the presence of the void. This is given mainly by the stress concentration effect of the void, which is most important for short cracks initiated from void. The influence on the stress intensity factors for the shorter cracks is more pronounced with higher void ratios *h_v_*/*r_v_*.

The influence of the void on the asymmetry of the cycle *R* is the same as in the case without the void—the asymmetry in modes II and III is not affected by the crack propagation, the asymmetry in mode I increases slightly with longer cracks.

The results of the third version of the model that contains the void and also includes friction between the crack faces are plotted in Figure 16. These figures illustrate the course of the stress intensity factors during the turn of the cylinder for the position *γ* = −90° and *γ* = 0°. The results from the model without friction are also plotted in these figures for comparison. The whole crack propagation in a rotating cylinder with a void and friction is broken down to 2D plots in the Appendix B—see Figure A7, Figure A8, Figure A9 and Figure A10.

The cycle of stress intensity factor *K_I_* has remained unchanged compared to the model with a void without friction. It is obvious, since the friction between crack faces cannot influence the tensile opening mode. The overall nature of the cycles of *K_II_* and *K_III_* with friction is also very similar to the previous model without friction. The friction affects mostly those values close to the positions of the closed crack—around the angle *ρ* = 90°. The difference between the values is approximately 5–10%. This means that friction does not significantly influence the overall *K_max_* functions, which can be seen also from the plot in Figure 17 that illustrates the overall influence of the friction on *K_IImax_* and *K_IIImax_* values during the crack propagation. There is again a slight shift in the positions where *K_II_* and *K_III_* cycles reach their maxima and minima with the growing crack (observable in Figure A7, Figure A8, Figure A9 and Figure A10).

It is important to remark here that heat is generated during friction, as the surfaces rub against each other. This effect is not considered in the described simulations. However, it may be significant, especially for cracks of longer lengths. Thermoplastic parts are generally not performing well in higher temperatures and even a slight increase in temperature can very negatively affect the resistance of the material against crack propagation. This is illustrated in the experimental study of mixed-mode I/III crack propagation in POM [31]. The work shows that the lifetime of the specimens decreases significantly with higher temperatures achieved as a result of friction in the vicinity of the crack. The high sensitivity of polymers to friction-generated heat is also reported in [32,40].

## 4. Conclusions

In the presented study, a parametrical model of a cylindrical bearing element with central crack made of the POM was developed. It was used to simulate and analyze the process of crack growth initiated from an internal defect, which is often the cause of failure of thermoplastic bearing elements.

The first version of the model contained just a circular flat discontinuity propagating from the middle of the cylinder towards its outer surface. It was observed that the gradual change in the orientation of the crack against the load during the rotation caused a combination of all three possible crack opening modes, characterized by stress intensity factors *K_I_*, *K_II_* and *K_III_*, on the crack front. Originally, it was assumed that the opening mode characterized by the *K_I_* was the dominant mode, but it was found that the crack was actually closed with the load pressing the faces together during most of the cycle. Additionally, the maximum values of the shear mode stress intensity factors, *K_II_* and *K_III_*, were higher than the maximum reached by *K_I_*.

A parametric study of the stress intensity factor values during crack propagation was carried out that resulted in the formation of parametric equations characterizing the course of the maximum values of stress intensity factors *K_Imax_*, *K_IImax_* and *K_IIImax_* during the rotation. These equations contain the dimensions of the cylinders and can be used to quickly describe the crack tip stress situation in the cylinder upon crack propagation without having to carry out a lengthy FEM simulation.

The second version of the model was made by adding a spheroidal void. The void simulated a presence of a manufacturing defect. It turned out that the presence of the void decreased the overall stiffness of the element and caused greater opening of the crack in mode I and, thus, higher *K_I_* in the position *γ* = 90° (and −90°). Contrary to that, the *K_I 0°_* and *K_II_* substantially decreased in the presence of the void. The *K_III_* remains almost unchanged even in cases with a rather large void. The common observation was that with the crack propagating further from the void, the void influence decreased up to a point where the stress intensity factor values were the same as for the flat crack.

The third version of the model included friction between the crack faces. It was found that the presence of friction did not substantially influence the values of stress intensity factor.

The potential of the results presented in this paper is in the possibility to use them for the calculation of lifetime estimations without complex numerical modelling. However, this requires reliable material data that characterizes the crack propagation rate under mixed-mode conditions.

## Figures and Tables

**Figure 1 materials-14-02656-f001:**
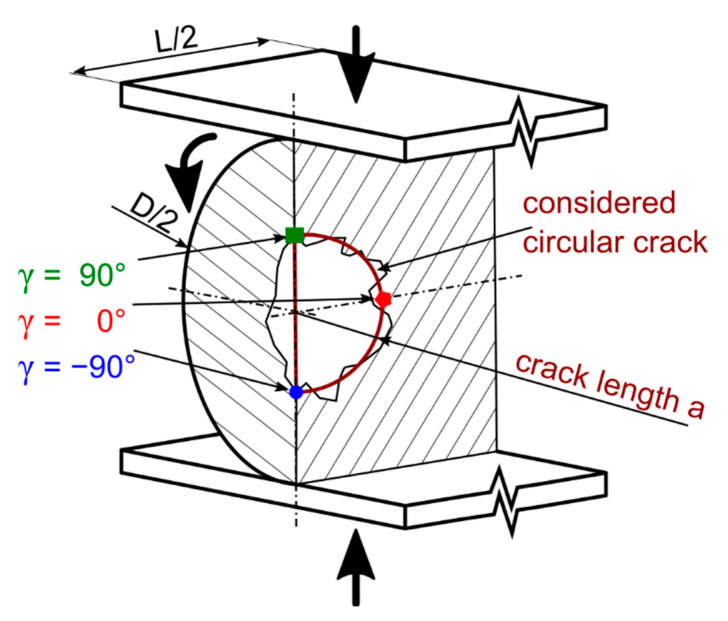
The situation of the bearing cylinder with an internal defect, compressed between two steel plates and rolling.

**Figure 2 materials-14-02656-f002:**
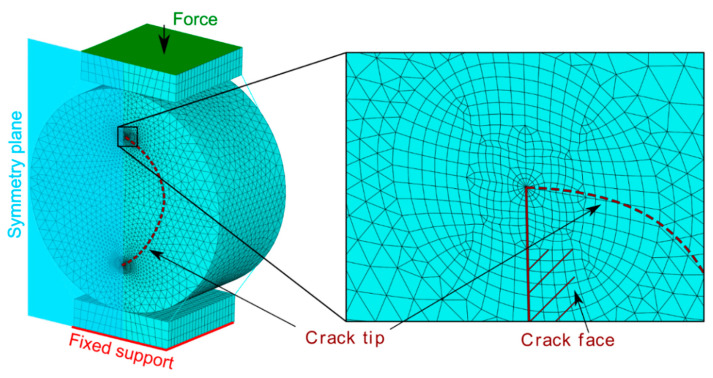
The mesh of the whole symmetrical FEM model of the bearing cylinder. The refined area in the vicinity of the crack tip is pictured in the detail view (note the special crack tip elements in the middle).

**Figure 3 materials-14-02656-f003:**
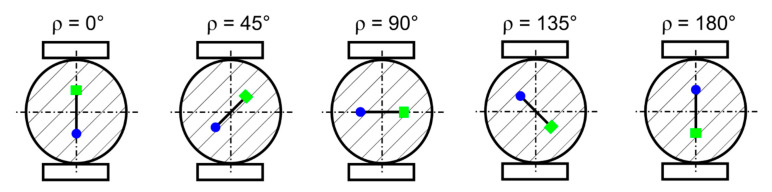
Schematic illustration of the changing orientation of the crack during rolling and the angle *ρ* that describes the rolling position.

**Figure 4 materials-14-02656-f004:**
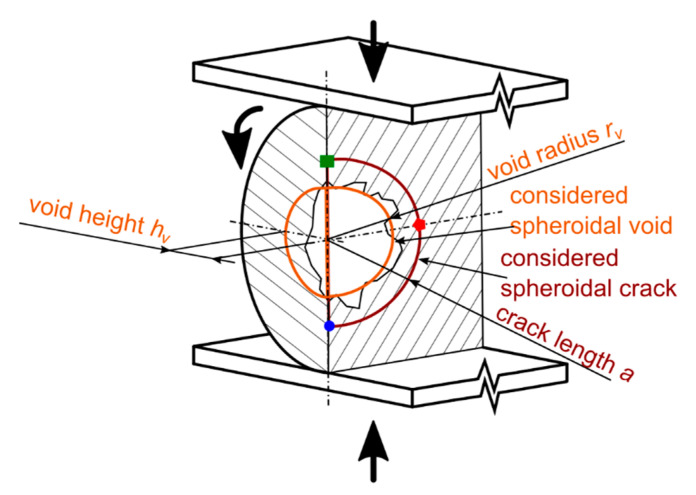
Model of a bearing cylinder with an initial spheroidal void in the middle.

**Figure 5 materials-14-02656-f005:**
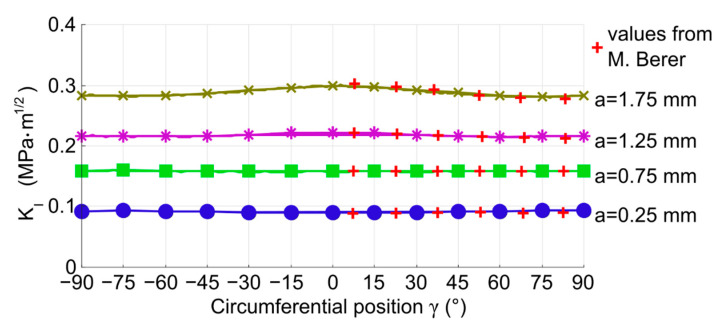
Comparison of *K_I_* values from Berer et al. [3] and from the newer model described in this paper. The values from the newer model were estimated by domain integral (solid lines) and node deformations (separate points).

**Figure 6 materials-14-02656-f006:**
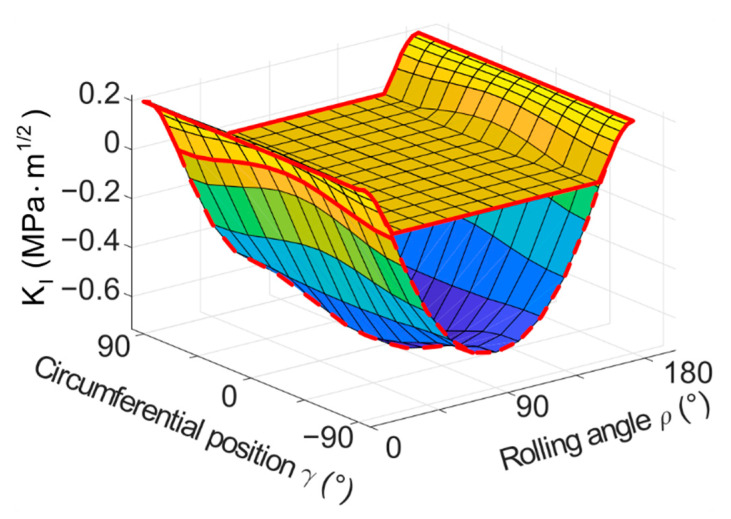
3D plot of *K_I_* as a function of both, the circumferential position on the crack front *γ* and the rolling orientation of the crack *ρ*, crack length *a* = 1.25 mm. The upper surface reflects the situation with contact defined between crack faces (highlighted by solid red line). The lower surface are theoretical negative values of *K_I_* acquired from a model where no contact was defined between crack faces (highlighted by dashed red line).

**Figure 7 materials-14-02656-f007:**
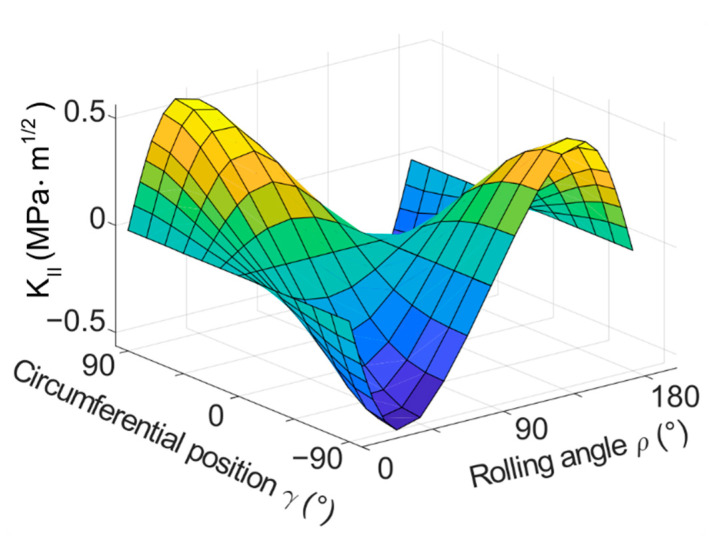
3D plot of *K_II_* as a function of both, the circumferential position on the crack front *γ* and the rolling orientation of the crack *ρ*, crack length *a* = 1.25 mm.

**Figure 8 materials-14-02656-f008:**
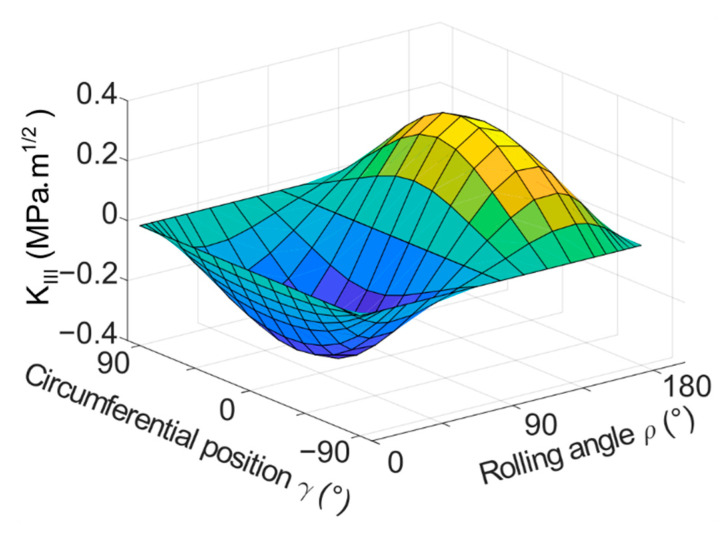
3D plot of *K_III_* as a function of both, the circumferential position on the crack front *γ* and the rolling orientation of the crack *ρ*, crack length *a* = 1.25 mm.

**Figure 9 materials-14-02656-f009:**
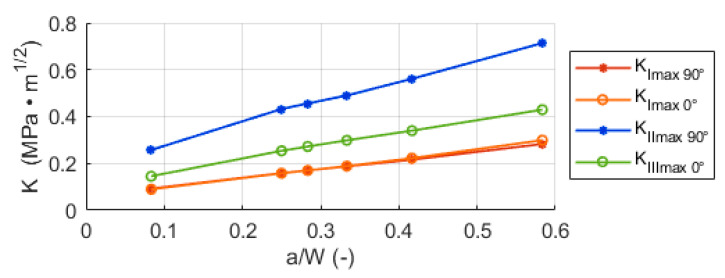
Maximum stress intensity factors in the rolling cycle for different crack tip positions *γ* depending on the normalized crack length. The indices 0° and 90° indicate the position *γ* at the crack front.

**Figure 10 materials-14-02656-f010:**
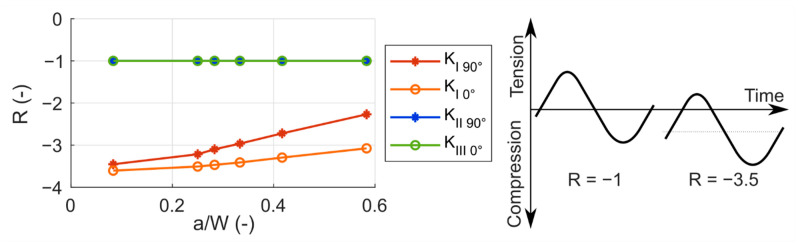
The *R*-ratio for different loading modes at different positions on the crack front during crack growth (**left**) with a schematic illustration of loading cycles with different *R*-ratios (**right**). The indices 0° and 90° indicate the position *γ* at the crack front.

**Figure 11 materials-14-02656-f011:**
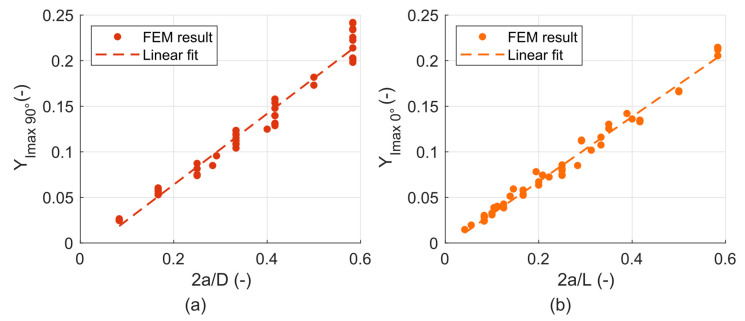
(**a**) Parametric functions fitting the values of *Y_Imax 90°_*; (**b**) Parametric functions fitting the values of *Y_Imax 0°_*.

**Figure 12 materials-14-02656-f012:**
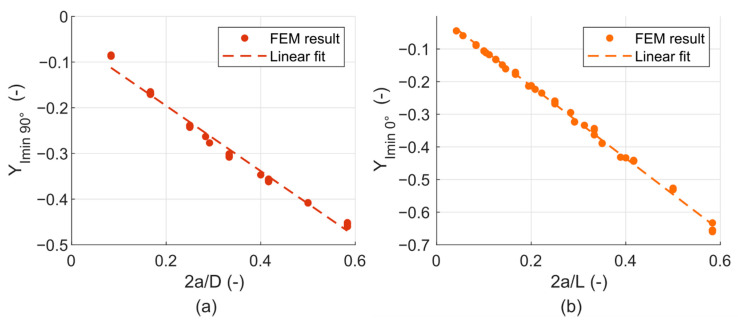
(**a**) Parametric functions fitting the values of *Y_Imin_*
_90°_; (**b**) Parametric functions fitting the values of *Y_Imin_*
_0°_.

**Figure 13 materials-14-02656-f013:**
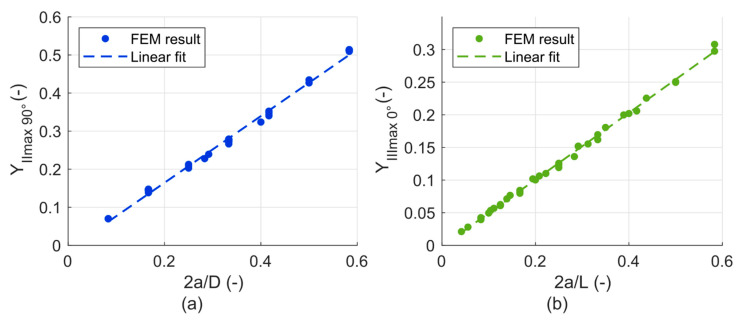
(**a**) Parametric functions fitting the values of *Y_IImax_*
_90°_; (**b**) Parametric functions fitting the values of *Y_IIImax_*
_0°_.

**Figure 14 materials-14-02656-f014:**
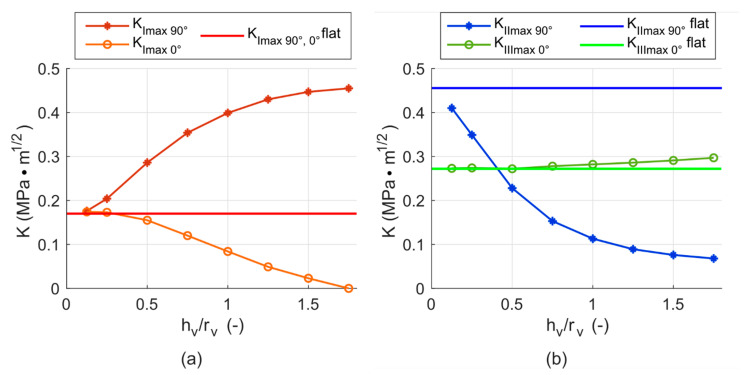
(**a**) Change of *K_Imax_* in different crack tip positions depending on the varying void ratio *h_v_*/*r_v_*; (**b**) Change of *K_IImax_* and *K_IIImax_* depending on the void ratio. Both plots contain the value of *K_Imax_*
_90°,0°_, *K_IImax_*
_90°_ and *K_IIImax_*
_0°_ for the case of flat crack *a* = 0.85 mm plotted as a solid line for comparison.

**Figure 15 materials-14-02656-f015:**
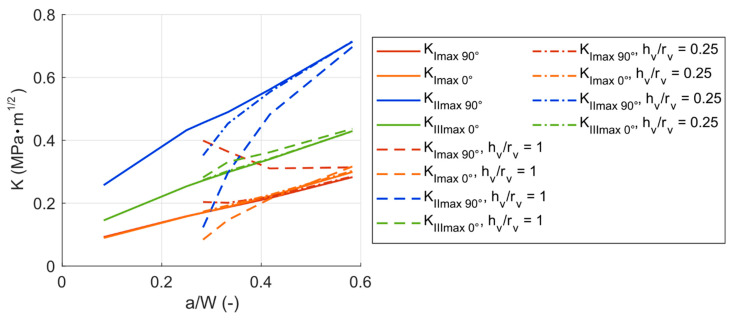
*K_max_* for different crack tip positions depending on the normalized crack length—comparison of the model with Table 0. mm, *h_v_/r_v_* = 0.25 or 1).

**Figure 16 materials-14-02656-f016:**
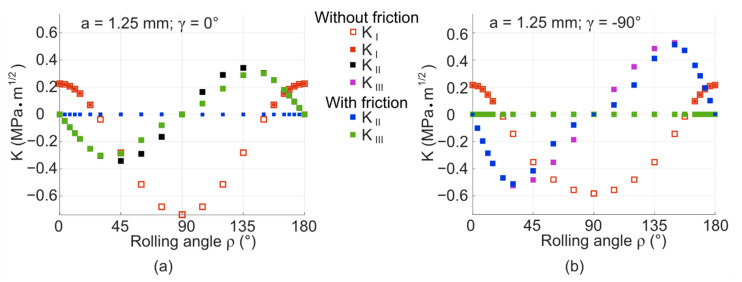
Comparison of stress intensity factor values for the model with void without friction and with friction: (**a**) *D* × *L* = 6 × 6 mm^2^, *a* = 1.25 mm; *γ* = 0°; (**b**) *D* × *L* = 6 × 6 mm^2^, *a* = 1.25 mm, *γ* = −90°.

**Figure 17 materials-14-02656-f017:**
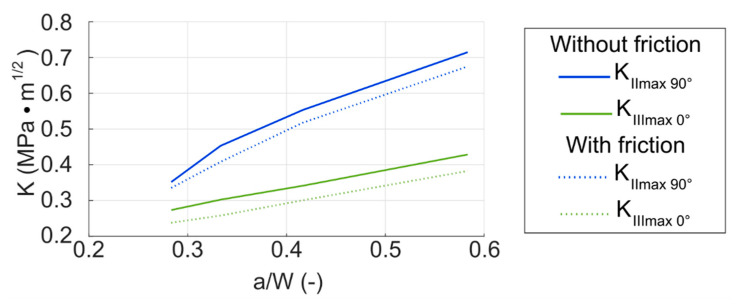
*K_IImax_* and *K_IIImax_* depending on the normalized crack length; comparison of model with void without friction and model with void with friction; void ratio *h_v_*/*r_v_* = 0.25, *r_v_* = 0.75 mm.

## Data Availability

The data presented in this study are available on request from the corresponding author.

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
