# Peer review of "Crack Propagation Analysis of Compression Loaded Rolling Elements"

_materials, 2021, doi:10.3390/ma14102656_

Round 1

Reviewer 1 Report

This manuscript investigated the effect of crack/void shape on the stress intensity factor for different case scenarios. Presented examples show the efficiency and accuracy of the approach. The manuscript covers a topic of relevance and is in the scope of the journal. However, the authors need to further clarify the following issues:

  1. Is it a compressive load or shear load, or combination of both? Figs. 1 and 4 showed shear load, while Fig. 2 showed a compressive one!
  2. Have you done any mesh sensitivity analysis?
  3. What is the reason to have a constant SIF while changing circumferential position?
  4. What type of friction model did you use in your simulation? Why
  5. Explain why SIF declines when there is friction in the model.
  6. Have you any type of crack propagation models in the ANSYS, or you just increased the crack size artificially?

Reviewer 2 Report

The problem of crack propagation from internal defects in thermoplastic cylindrical bearing elements is addressed in this paper. The contents of current work should be of interest for the field. However, the following comments should be addressed.

  1. In introduction, when mentioning fatigue failure analysis under production defects, more recent progress on this should be included, like “Int J Fatigue, 2019, 126: 165-173” and “Int J Fatigue, 2021, 146: 106157”;
  2. In section 2.1, “a circular crack” is considered, more text or references can be added to introduce that how to set a crack on the 3D model of ANSYS.
  3. In section 2.1, it writes “Quadratic tetrahedral elements were used to mesh the most of the model.”, why not use the regular quadratic bricks same with the steel plates.
  4. On page 4, “0.85 mm and 1.00 mm” and “3.8°, 7.5°, 11.2° and 22.5° as well as 157.5°, 168.8°, 172.5° and 176.2.” are added, the reviewer is curious about the effects and sources of values.
  5. The final FEM figures are more meaningful, which can be shown in the work.
  6. In figure 6, it strange that there are two surfaces, namely “3D plot of KI as a function”.
  7. The results and discussions are good, but the final conclusions should be refined and shortened, just focus on the main points of current work.

Round 2

Reviewer 1 Report

The Authors have addressed this reviewer's concerns and questions in a satisfactory way, and I would like to thank them again for their efforts.

Reviewer 2 Report

The work has been well refiend, it can be accepted as it is.